# Effectiveness of Nitrate Intake on Recovery from Exercise-Related Fatigue: A Systematic Review

**DOI:** 10.3390/ijerph191912021

**Published:** 2022-09-23

**Authors:** José M. Gamonales, Daniel Rojas-Valverde, Jesús Muñoz-Jiménez, Walter Serrano-Moreno, Sergio J. Ibáñez

**Affiliations:** 1Research Group in Optimization of Training and Sports Performance (GOERD), University of Extremadura, Av. De la Universidad, s/n, 10003 Cáceres, Spain; 2Facultad de Ciencias de la Salud, Universidad Francisco de Vitoria, 28223 Pozuelo de Alarcón, Spain; 3Núcleo de Estudios Para el Alto Rendimiento y la Salud (NARS-CIDISAD), Escuela Ciencia del Movimiento Humano y Calidad de Vida (CIEMHCAVI), Universidad Nacional, Heredia 863000, Costa Rica; 4Clínica de Lesiones Deportivas (Rehab&Readapt), Escuela Ciencia del Movimiento Humano y Calidad de Vida (CIEMHCAVI), Universidad Nacional, Heredia 863000, Costa Rica; 5Posgrado en Ciencias Médicas, Facultad de Medicina, Universidad de Colima, Colima 28040, Mexico

**Keywords:** supplement, anaerobic, beetroot, performance, training

## Abstract

Background: Recovery between efforts is critical to achieving optimal physical and sports performance. In this sense, many nutritional supplements that have been proven to improve recovery and physical and physiological performance are widely used. Supplements such as nitrates (NO_3_^−^), including organic foods such as beets, promote muscle recovery and relieve fatigue. This study aimed to comprehensively summarise the available literature on the effect of NO_3_^−^ consumption on exercise-related fatigue and muscle damage. Methods: A systematic search was carried out based on the Preferred Reporting Items for Systematic Reviews and Meta-analyses (PRISMA) using electronic databases (e.g., PubMed, Scopus, and Web of Science). From a total of 1634 studies identified, 15 studies were included in this review. Results: Based on the review, NO_3_^−^ intake provokes physiological and metabolic responses that could potentially boost exercise-related recovery. NO_3_^−^ could improve recovery indicators related to strength, pain, inflammation, and muscle damage. Conclusions: Despite the relative proven effectiveness of NO_3_^−^ on recovery after aerobic and anaerobic efforts, based on the heterogeneity of the procedures (e.g., dosage, chronic vs. acute intake, participants’ characteristics, variables and outcomes), it could be premature to suggest its extended use in sports.

## 1. Introduction

Damage and fatigue recovery plays a critical role in sports performance, considering that the sporting result is usually defined by increasingly smaller margins [1]. Although training promotes repeatedly tolerating great effort, the ability to recover between these efforts stands out in elite competitions due to its essential role in the training-adaptation cycle [2]. In this sense, a large number of dietary supplements, such as caffeine, beta-alanine, sodium bicarbonate, and magnesium, have proven their ability to improve performance and recovery [3].

In addition, the intake of organic products has stood out due to their benefits in reducing muscle pain and stiffness, lowering the perception of pain, decreasing the severity of an injury, modulating proinflammatory cytokines and anti-oxidative capacity, and reducing the burden of the gastrointestinal tract [4]. This is the case with pomegranate [5], green tea [6], cherry [7], turmeric [8], spinach [9], and beetroot [10].

Specifically, the acute [11] and chronic [12] intake of beetroot-derived products, which contain a large number of nitrates (NO_3_^−^), allows a potential improvement in the recovery of exercise-related fatigue. NO_3_^−^ has recently been studied in exercise and sports sciences [13] due to its vasodilator properties, improved oxygen availability, exercise tolerance, reduced cardiac output, vagal heart rate control recovery, and decreased muscle pain, among others [10,14,15].

Due to the entero-salivary cycle, NO_3_^−^ can be sequentially reduced to nitrites and subsequently to nitric oxide (NO); the latter causes regulatory effects at the vascular, metabolic, and immune levels [16]. Recent studies have proposed that NO may exert anti-inflammatory effects by hindering leukocyte activation and subsequent reduction of pro-inflammatory mediators that saturate the cell [17]. Additionally, regulating blood flow and muscle contractility influences glucose and calcium homeostasis, causes biogenesis, collaborates with mitochondrial respiration, and reduces the cost of O_2_ [18,19].

The rate of O_2_ availability is key to recovery from fatigue and increased mitochondrial efficiency. Both processes can be enhanced after NO_3_^−^ consumption due to increased blood flow and better adaptation of local perfusion to the metabolic rate during and after exercise [20]. Additionally, these benefits could impact the recovery rate and availability of energy reserves such as muscle glucose and phosphocreatine [21].

To date, little research has been carried out on the effects of nitrates on recovery from exercise-related fatigue. Despite this, the properties demonstrated concerning the improvement in mitochondrial respiration and O_2_ metabolism, as well as the anti-inflammatory and antioxidant properties, present a promising scenario in the consumption of NO_3_^−^ to counteract the effects of post-physical exertion fatigue. For this reason, this systematic review aims to systematise the scientific knowledge reported to date concerning the impact of NO_3_^−^ consumption on exercise-related fatigue recovery factors. The information from this study could help develop new NO_3_^−^ intake strategies and clarify the scenarios and conditions in which this consumption is effective and efficient in recovering physical and physiological capacities.

## 2. Materials and Methods

### 2.1. Study Design

This systematic literature review was prepared following the Preferred Reporting Items for Systematic Reviews and Meta-analyses (PRISMA) guidelines [22]. The authors selected the inclusion criteria and included experimental and quasi-experimental studies exploring the effectiveness of nitrate intake on recovery from exercise-related fatigue. The decision to perform a systematic review related to the aim of the present study is based on clear guidelines for the replication and updating of systematic reviews [23,24].

### 2.2. Selection Criteria

For this systematic review, the following inclusion criteria were established: (1) the study must determine the effect of NO_3_^−^ consumption on recovery from physical exercise; (2) studies with an experimental or quasi-experimental design; (3) studies with pre-and post-intervention assessments; (4) studies published in the English language or that allow its translation; and (5) studies published from 2000 to 2022 (15 June).

On the other hand, studies were excluded if: (1) the full text was not available; (2) the measurement protocol and key methodological aspects of food consumption related to NO_3_^−^ were not specified (e.g., dosage, exercise performed); (3) fatiguing stimuli other than physical exercise were used; (4) studies carried out in people with some pre-existing pathology; (5) studies that do not correspond to original experiments (e.g., editorials, opinion, reviews), (6) studies analysing the effect of NO_3_^−^ intake in performance, (7) studies analysing fatigue during exercise and no during recovery period; and (8) studies that consider other methods of recovery after physical exercise and studies the effects of NO_3_^−^ consumption on performance during exercise.

### 2.3. Data Sources and Search Profile

The electronic databases PubMed (MedLine), Science Direct (Scopus), and Web of Science (WoS) were chosen to perform a literature search according to the research topic. Other secondary studies were obtained from alternative databases manually. This search was limited from 2000 to 15 June 2022. To search for results, a combination of keywords such as “Nitrates” OR “Beetroot” OR “Beta vulgaris” OR “Beet” OR “Spinach” AND “Recovery” AND “Exercise”; were used. The Boolean AND operator were used to combine the keywords. In Figure 1, the flow diagram of the search process is shown.

### 2.4. Studies Selection and Data Extraction

The main author carried out the initial search. A database was created in a computer program (Excel 16.6, Microsoft, CA, USA), which included each document found in each database. In this database, the name of the database, the title of the article, the authors, the name of the journal in which it was published, and the year of publication was included. Subsequently, duplicate manuscripts were eliminated, and the title and abstract of the remaining documents were read. 

The full text was read to verify that the proposed eligibility criteria are met to judge the article’s relevance. The data extracted from the manuscripts were similar to previous systematic reviews on the topic of the use of nitrates in sports as follows [10,14]: author and year, characteristics of participants, study design, fatiguing stimulus, the dosage of nitrates, results in the percentage of improvement, and effects of supplementation.

## 3. Results

As shown in Figure 1, after evaluating 1634 identified studies, 1065 were excluded based on title or abstract, 497 due to the year of publication, and 25 by duplicity. The remaining 47 studies were examined in full text, of which 15 articles fulfilled the inclusion criteria. Studies were excluded after full-text reading because of no clear information on the characteristics of the athletes, the main objective was analysing the effect on performance, studies in other languages, studies not using NO_3_^−^, and studies nots exploring the impact of NO_3_^−^ on recovery-related variables. The total documents included in this systematic review are shown in Table 1. The studies explored the effect in male (*n* = 305, 93.3%) and female (*n* = 22, 6.7%) participants. The sample of this systematic review was mostly runners (*n* = 147, 46.7%), followed by active healthy participants (*n* = 104, 33.0%), triathletes (*n* = 32, 10.2%), military personnel (*n* = 22, 7%), football players (*n* = 13, 4.1%) and cyclists (*n* = 9, 2.9%).

Concerning the studies’ design, most of the evidence was based on double-blinded, placebo and randomized-controlled trials (*n* = 9, 64.3%), while the remaining studies were single-blinded (*n* = 6, 40.0%). The authors selected one, two, and three groups. Assigning the participants to groups in beetroot supplementation (*n* = 11, 78.6%), sodium nitrate intake (*n* = 2, 13.3%), antioxidants intake (*n* = 1, 6.67%), citrulline supplementation (*n* = 1, 6.67%), or placebo (*n* = 15, 100%) as control group. Only one study used two different nitrate dosages for comparison.

The fatigue stimuli varied significantly among aerobic, anaerobic, and intermittent exercise. The most used stimuli were marathon running (*n* = 3, 20.0%), strength training (*n* = 3, 20.0%), drop jumps (*n* = 2, 13.3%), and aerobic cycling in ergometer (*n* = 2, 13.3%), and only one study used the following activities: triathlon, military expedition, uphill walking, repeated sprints, and intermittent shuttle test.

The dosage used in the different studies differed by the product presentation, capsules or pills, drinks, and gels. In addition, there were acute (*n* = 6, 40.0%) and chronic (*n* = 10, 66.7%) intake of nitrates. The chronic intake was made during 1, 2, 7, 8, 11, and 9 weeks. Moreover, only two (13.34%) studies performed an intake after exercise, in contrast to 14 (93.3%) studies performing the NO_3_^−^ intake before exercise. The acute intake ranged from 2 h to 24 h before the exercise.

The variables used to identify the effectiveness of recovery of the NO_3_^−^ intake varied from metabolites (e.g., glycerol, arabitol, xylose, α-Oleoylglycerol), strength and power performance (e.g., maximal isometric voluntary contraction, countermovement jump), cardiovascular indicators (e.g., heart rate, blood pressure), muscle damage biomarkers (e.g., creatine kinase, lactate dehydrogenase), inflammatory response cytokines (e.g., IL-6, IL-8, reactive C-protein, reactive oxygen species), and other perceptual pain variables (e.g., delayed onset muscle soreness, pain pressure threshold).

A total of nine studies (*n* = 10, 66.7%) reported positive results from the intake of NO_3_^−^ in variables such as maximal isometric voluntary contraction, recovery heart rate, cortisol levels, blood pressure, countermovement jump, squat jump, antioxidants levels, and pain pressure threshold. Among the activities that reported positive results when using NO_3_^−^ were those with anaerobic and intermittent components (*n* = 5, drop jumps, strength training, shuttle test) and those of aerobic nature (*n* = 4, triathlon, marathon, uphill walking, military expedition).

## 4. Discussion

The present study aims to systematise the scientific knowledge reported to date concerning the effect of NO_3_^−^ consumption on exercise-related fatigue recovery factors to develop new strategies regarding NO_3_^−^ intake. The evidence published related to the aim of the study is relatively scarce and is characterised by its divergence. Furthermore, the procedure used in the systematic review of the manuscripts has followed a similar procedure reported in the scientific literature [22] regarding selecting the most suitable research-topic documents. Likewise, the employed method in the revision has been used for different authors from different scopes [10,38,39]. Moreover, this systematic review identified that more than 60% of the included research reported positive effects on recovery indicators due to the intake of NO_3_^−^. However, due to the heterogeneity of consumption doses, subject characteristics, and fatiguing stimuli and variables used, it is premature to indicate that there is real effectiveness of NO_3_^−^ consumption in recovery in sports and exercise.

Because exercise-induced fatigue and muscle damage can cause significant discomfort and impair functionality and performance, strategies must be used to recover between efforts as a key phase in sports programming. In this sense, NO_3_^−^ intake provides a series of benefits and causes a physiological cascade that promotes recovery by creating a series of anti-inflammatory, antioxidant, and biogenesis responses, among others [26].

Concerning the cardiovascular responses to NO_3_^−^ intake, several studies report positive benefits of NO_3_^−^ such as s lowering blood pressure [40,41], angiogenesis, and improving muscle perfusion. The vasodilation caused by the consumption of NO_3_^−^ leads to a decrease in heart rate and blood pressure, as was found in the systematic review. These responses improve tissue perfusion, including muscle during exercise and recovery [42]. Considering that blood pressure results from the cardiac output and total peripheral resistance, the improvement in endothelial function allows for increased vascular compliance [43]. This is a sign that the autonomic nervous system is functioning well and is returning to a normal state [44].

Another benefit of NO_3_^−^ consumption is the improved muscle contractile efficiency [41] due to the leakage promotion of calcium from the sarcoplasmic reticulum, muscle oxygenation and consumption [45], and finally leading to improved performance [41,46,47]. Moreover, NO_3_^−^ intake could lead to increased mitochondrial proteins [45], meaning that not only can a better neuromuscular function be achieved after exercise, but also protein resynthesis and, therefore, the recovery of contractile capacity could improve. This is evidenced in the present study due to increased jumping capacity and isometric contractility.

Other indicators of recovery are perceptual. In this review, it was found that reported muscle pain, delayed onset muscle soreness, and pain pressure threshold improved with NO_3_^−^ consumption. This finding is particularly important due to perceptual variables’ role in recovery and performance [48,49]. Pain is strongly related to muscular contraction. Exercise-induced pain is related to feedback from nociceptive group muscle afferents about alteration associated with muscle function as increased intramuscular pressure, heat, high levels of metabolites, and deformation of tissues [34]. Due to the scarce evidence regarding the relationship between the consumption of nitrates and the perception of pain, it should be clarified if the recovery comes from the reabsorption of metabolites. Therefore, we must determine whether it is the decrease in perception of pain or from the blockage of nociceptive afference that allows an athlete to exert greater effort or withstand greater loads for a longer period. Future research must focus on the nociceptive physiological implications of NO_3_^−^ consumption.

This study suggested that no markers of exercise recovery were significantly enhanced by NO_3_^−^ intake. Some well-known biomarkers [50] such as creatine kinase, lactate dehydrogenase, C-reactive protein, and other proinflammatory cytokines (e.g., IL-8, IL-6), remain unaltered by NO_3_^−^. In contrast, cortisol decreased in a study. This biomarker suggests chronic stress, and its elevation could prolong recovery and potentially open illness susceptibility [51]. A more in-depth analysis is required to establish a link between NO_3_^−^ and cortisol regulation to understand its effects on recovery via this supplement consumption.

Finally, about the dose, recent studies have shown that high doses (8–16 mmol) of NO_3_^−^ help raise blood levels of desired NO to cause the effects of consumption of this supplement at a lower level [10]. Based on the results of this systematic review, the dosage would depend on the duration of the administration of NO_3_^−^. Some differences between acute and chronic supplementations were found. In recovery, acute intakes are characterised by relatively high doses, varying between 12.9 mmol to 20.5 mmol of NO_3_^−^. In contrast, chronic intakes usually range from 3.4 mmol to 12.5 mmol of NO_3_^−^. The above will also depend on the form of consumption and the athlete’s tolerance.

Consequently, due to its relatively low consumption, the chronic dose should be administered at least twice a day for 4–6 days [52,53,54,55,56], as reported in this review (consumption for 7–11 days and up to 9 weeks). On the other hand, acute intake must be administrated two to three hours before training or competition in a single dose of approximately 5 mmol to 6.5 mmol of NO_3_^−^ [18,57,58]. In this review, the acute administration was higher than the recommended performance; studies that use the post-exertion dose for recovery are systematised for the first time, as is common for these objectives in sports [59,60].

Despite there being NO_3_^−^ dosage recommendations, based on the presented evidence, there is a significant limitation of the studies regarding the supplementation of NO_3_^−^. Considering that other well-known dietary supplement guidelines are usually prescribed relative to the athlete’s weight [61,62], the used dosage in the studies included in this systematic review usually corresponds to a commercial product instead of the athlete’s needs. Consequently, the NO_3_^−^ should be intake per kilogram of weight instead of a single homogeneous dose.

### 4.1. Limitations

While this study presents some evidence on the NO_3_^−^ intake effects on exercise-related recovery indicators, these findings must be seen as having some limitations. Due to the wide range and heterogeneity of the participants’ characteristics, exercising protocols, and dosage used, it is difficult to fully interpret the results of the studies. These differences between the variables used to prove the effectiveness of recovery varied significantly. Therefore, this issue did not allow us to apply some statistical analysis (e.g., effect sizes) to see a cumulative effect throughout the manuscripts. Clear differences in the distances, times, type of exercise, and participant’s fitness level make it difficult to compare results and report the real effect of BRJ on exercise-related recovery. In addition, there were some difficulties in interpreting the actual impact of NO_3_^−^ intake due to the varied design protocols of the studies. The consumption of beet juice has direct effects on the body, one of which is the change of urine colour [10]; this is an aspect that generates complexity in applying double-blind trials.

### 4.2. Future Research Considerations

Future studies should consider a larger number of participants. Further information is required regarding the potency of these intake strategies in female populations and athletes to explore the efficacy of NO_3_^−^ supplements. Evidence suggests that fitness and training status could impact dietary supplementation [63].

Future studies could explore the combined effect of NO_3_^−^ with multiple nutritional or supplementation strategies on recovery-related indicators, considering antioxidants and citrulline showed good results in this review. In addition, more information is required regarding the interplay between the supplementation strategies of NO_3_^−^ and other recovery options such as rest, manual therapy, or cryotherapy.

Greater efforts should be made concerning the optimal dose recommended for NO_3_^−^ consumption. It is critical to study the possible health consequences of its long-term consumption in large quantities [64]. This would allow generating an alert to athletes or not recommending the consumption of NO_3_^−^ in those cases where the evidence does not indicate an ergogenic effect for recovery. For this, the customisation of the doses is key and should be explored in future research.

### 4.3. Practical Applications

Considering the limitations indicated above, it can be recommended to use the NO_3_^−^ in aerobic and anaerobic sports, acutely or chronically. When an acute intake strategy is used, intakes should be relatively high (12–20 mol of NO_3_^−^), and when a chronic intake strategy is used, it should be characterised by low intakes (3.5–12 mmol of NO_3_^−^). Despite these basic recommendations, the nutritionist, coach, and athlete must understand that the optimal dose will lie in personalising it, making the necessary adaptations according to the particularities of the athlete and the discipline he practices. Additionally, one must be very careful with prescribing these supplements for recovery in those populations that have not been studied, such as women, elite athletes, and older adults, among others who exercise or play sports. In this sense, there are some latent risks if NO_3_^−^ daily requirements are exceeded [15,65,66]. Additional considerations regarding dosage, intake duration, and wash-up, and some supplementation timing factors should be considered when prescribing this kind of recovery aid.

The above is because the contact of NO_3_^−^ with anaerobic bacteria in the saliva is ne-cessary to convert it to NO_2_^−^ [65]. Other precautions should be taken; for example, when using NO_3_^−^, some mouthwashes or other similar substances must be avoided during the consumption since it may lead to difficulties in NO_3_^−^ absorption process. In addition, NO_3_^−^ based products are usually sweet and have an earthy flavor, which might not be to the athletes’ liking.

## 5. Conclusions

NO_3_^−^ intake could provoke a series of physiological responses that suggest great potential in its use for exercise-related recovery and are consistent with this study’s results. These include vasodilation, decreased blood pressure and heart rate, regulation of blood flow, improvement of muscle and mitochondrial contractility and oxygenation, glucose and calcium homeostasis, biogenesis, and regulation of cortisol and testosterone, among others.

The effects on recovery of the NO_3_^−^ dosing may vary depending on intake duration (e.g., chronic or acute), type of exercise (e.g., aerobic vs. intermittent vs. anaerobic), and muscle contraction (e.g., eccentric vs. concentric) and fatigue-related indicators such as reference (e.g., biochemical vs. performance testing vs. cardiovascular) and timing (e.g., before, during, after exercise) among other factors.

Despite more than 60% of the studies presenting significant results when consuming NO_3_^−^, due to the wide range of participants’ characteristics, study design, dosing, and outcomes variables, more studies are required to explore these possibilities and variants in the consumption of this supplement to indicate its effectiveness safely.

Future research may dig deeper into the effect of extended NO_3_^−^ intake protocols, higher dosing, diverse sports disciplines and modalities, and combination with other su-pplements. Considering the impact of NO_3_^−^ on recovery, it could be interesting to explore the effects of its intake on inflammation, muscle damage, central and peripheral fatigue, and as a recovery aid. This research should be developed considering the potential risks of exceeding acceptable NO_3_^−^ daily intake.

## Figures and Tables

**Figure 1 ijerph-19-12021-f001:**
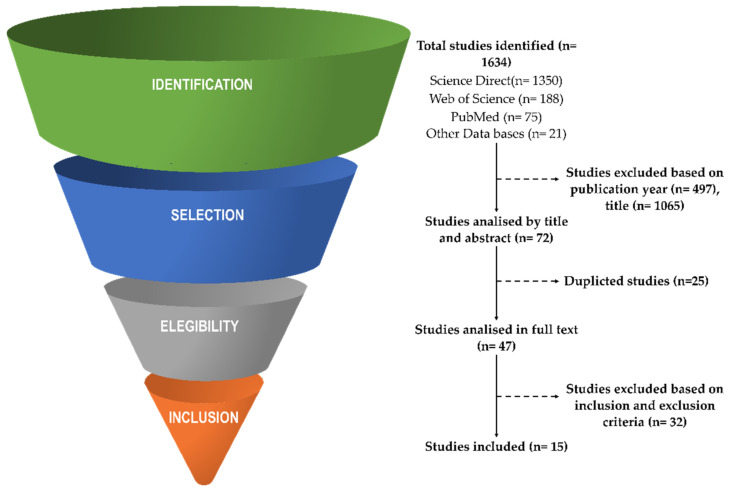
Flowchart of the identification, selection, and discrimination of articles.

**Table 1 ijerph-19-12021-t001:** Data extracted from the included studies.

#	Author/Year	Participants	Study Design	Fatigue Stimulus	NO_3_^−^ Dosage	Variables	Outcomes
1	Clifford, Bell et al., (2016) [25]	*n* = 30, ♂ Active	Double-blind Placebo Randomized-Controlled-Trial (BRJ_1_ vs. BRJ_2_ vs. PLA)	100 drop jumps	BRJ_1_ = 4.0 mmol/NO_3_^−^ (Chronic: 2–3 doses/3 days, 7 total) BRJ_2_ = 4.0 mmol/NO_3_^−^ (Chronic: 2–3 doses/3 days, 7 total)	MIVC CMJ PPT CK IL-6 IL-8 TNF-α	BRJ_1_ and BRJ_2_ enhance CMJ and PPT recovery
2	Clifford et al., (2016) [26]	*n* = 20, ♂ Active	Double-blind Placebo Randomized-Controlled-Trial (BRJ vs. PLA)	2× repeated sprint test (20 × 30 m sprints: 30 s recovery)	BRJ = 4.0 mmol/NO_3_^−^ (Chronic: 2–3 doses/3 days, 8 total)	MIVC CMJ PPT Reactive strength CK CRP PC	BRJ enhances CMJ and PPT recovery
3	Clifford, Allerton et al., (2017) [27]	*n* = 34, ♂ Runners	Double-blind Placebo Randomized-Controlled-Trial (BRJ vs. PLA)	Marathon	BRJ = 3.4 mmol/NO_3_^−^ (Chronic: 2–3 doses/3 days, 7 total)	MIVC CMJ CK DOMS CRP IL-6 IL-8 TNF-α	No differences found
4	Clifford, Howatson et al., (2017) [28]	*n* = 30, ♂ Active	Double-blind Placebo Randomized-Controlled-Trial (BRJ vs. NaNO_3_^−^ vs. PLA)	100 drop jumps	BRJ = 3.4 mmol/NO_3_^−^ (Chronic: 2–3 doses/3 days, 7 total) NaNO_3_^−^ = 3.4 mmol/NO_3_^−^ (Chronic: 2–3 doses/3 days, 7 total)	MIVC CMJ PPT CK CRP	BRJ enhance PPT recovery
5	Clifford et al., (2018) [29]	*n* = 30, ♂ Runners	Double-blind Placebo Randomized-Controlled-Trial (BRJ vs. PLA)	Marathon	BRJ = 4.0 mmol/NO_3_^−^ (Chronic: 2–3 doses/6 days, 11 total)	ROS mtDNA damage	No differences found
6	Carriker et al., (2018) [30]	*n* = 9, ♂ Cyclists	Double-blind Placebo Randomized-Controlled-Trial (BRJ vs. PLA)	25–70% of normobaric VO2max in a hypobaric chamber at 3500 m (5 min in, 4 min off)	BRJ = 12.8 mmol/NO_3_^−^ (Acute: 2.5 h pre-stimuli)	8-isoprostane Catalase	No differences found
7	Waldron et al., (2018) [31]	*n* = 8, ♂ Active	Placebo Randomized-Controlled-Trial (BRJ vs. PLA)	Intermittent walking at 3 km/h with gradients between 1–20%	BRJ = 350 mL (20.5 mmol/NO_3_^−^) (acute: 24 h before exercise)	Heart rate VO_2_ Blood pressure Glucose Potassium Blood lactate	BRJ enhances heart rate, blood pressure and VO_2_ recovery
8	Larsen et al., (2019) [32]	*n* = 30, ♂ Active	Placebo Randomized-Controlled-Trial (BRJ vs. Antioxidants vs. PLA)	10× Eccentric dorsiflexion: 30 s rest (until volitional fatigue)	BRJ = 12.9 mmol/NO_3_^−^ (Acute: 1 × 48 h post-exercise) Antioxidants = No specified, cocktail shot (Acute: 1 × 48 h post-exercise)	MIVC PPT DOMS	No differences found
9	Menezes et al., (2019) [33]	*n* = 14, ♂ Active	Placebo Randomized-Controlled-Trial (NaNO_3_^−^ vs. PLA)	30 min cycling (50% max power)	NaNO_3_^−^ = 10 mg/kg (body weight) (Acute:chronic [5 days])	Ferric reducing antioxidant power Uric acid Superoxide Thiobarbituric acid	NaNO_3_^−^ enhances an increase in Ferric, reducing antioxidant power and uric acid and decrease of superoxide and thiobarbituric acid
10	Husmann et al., (2019) [34]	*n* = 12, ♂ Active	Double-blind Placebo Randomized-Controlled-Trial (BRJ vs. PLA)	One-leg dynamic isotonic contractions	BRJ = 70 mL, 6.5 mmol/NO_3_^−^ (Chronic: 5 days before exercise)	MVC Rating of perceived effort Leg muscle pain Peripheral nerve stimulation	BRJ enhances muscle contraction function and pain and effort perception.
11	Marshall et al., (2021) [35]	*n* = 22, ♂ = 12, ♀ = 10 Military	Placebo Randomized-Controlled-Trial (BRJ vs. PLA)	High Altitude military expedition	BRJ = 70 mL, 12.5 mmol/NO_3_^−^ (Chronic: 1 dose/day, 11 total)	SO_2_ Heart Rate Diastolic Pressure Rate Perceived exertion High altitude illness	BRJ enhances heart rate recovery speed
12	Daab et al., (2021) [12]	*n* = 13, ♂ Footballers	Double-blind Placebo Randomized-controlled-Trial (BRJ vs. PLA)	Intermittent Shuttle Test	BRJ = 4.0 mmol/NO_3_^−^ (Chronic: 2 doses/day, 14 total)	MIVC CMJ Squat Jump 20 m sprint DOMS CK CRP	BRJ enhances MIVC, CMJ, SJ, 20 min sprint, and DOMS recovery
13	Stander et al., (2021) [36]	*n* = 31, ♂ = 19, ♀ = 12 Runners	Placebo Randomized-controlled-Trial (BRJ vs. PLA)	Marathon	BRJ = 3.4 mmol/NO_3_^−^ (immediately post-marathon: 3 × 250 mL, 1st day post-marathon: 3 × 250 mL, 2nd day post-marathon: 1 × 250 mL) CTRL = maltodextrin, protein powder and fruit squash	Metabolites (e.g., glycerol, arabitol, xylose, α-Oleoylglycerol)	No differences found
14	Benjamim et al., (2021) [11]	*n* = 12, ♂ Active	Double-blind Placebo Randomized-controlled-Trial (BRJ vs. PLA)	75% 1RM strength exercise	BRJ = 600 mg/NO_3_^−^ (acute: 120 min before exercise)	Heart rate Heart rate variability Blood pressure	BRJ enhances heart rate and systolic blood pressure recovery
15	Burgos et al., (2022) [37]	*n* = 32, ♂ Triathletes	Double-blind Placebo Randomized-controlled-Trial (BRJ vs. Citrulline vs. BRJ + Citrulline vs. PLA)	135 h training	BRJ = 100 mg/NO_3_^−^ (chronic: 9 weeks) Citrulline = 3 g/day (chronic: 9 weeks)	Urea Creatinine CK Lactate dehydrogenase Testosterone Cortisol Cooper test	BRJ + Citrulline enhance recovery by a reduction in cortisol and an increase in testosterone/cortisol ratio.

BRJ = beet root juice; PLA = placebo; MIVC = maximal isometric voluntary contraction; MVC, maximal voluntary contraction; CMJ = countermovement jump; PPT = pain pressure threshold; CK = creatin kinase; IL = interlukine; PC = phosphocreatine; DOMS = delayed onset muscle soreness; CRP = C-reactive protein; NO_3_^−^ = nitrates; NaNO_3_^−^ = sodium nitrates; ROS = reactive oxygen species; SO_2_ = oxygen saturation.

## Data Availability

Not applicable.

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
