# Peer review of "Effectiveness of Nitrate Intake on Recovery from Exercise-Related Fatigue: A Systematic Review"

_ijerph, 2022, doi:10.3390/ijerph191912021_

Round 1

Reviewer 1 Report

This is a systematic review of the effectiveness of nitrate supplementation on exercise-related fatigue. 

Overall the paper is well written and concise. The table was well thought-out and was easy to read. The discussion summarized the table and discussed the limitations, practical applications, and main points. 

This was a novel approach to reviewing nitrate supplementation. There are multiple reviews and meta-analyses reviewing performance. This seems like the first to review fatigue. 

Below are a few suggestions to strengthen the paper. 

Line 23: I don't think focusing on "organic" adds to the paper. Almost all supplementation is organic. This was not developed through the paper and I think it is assumed.

When referencing the studies performed with performance in the intro you might want to reference the 2021 review & meta-analysis from MSSE (DOI: 10.1249/MSS.0000000000002363), it seemed like it fit your criteria. 

You may want to consider mental and cognitive fatigue as well, 2014 Thompson (doi: 10.1016/j.resp.2013.12.015)

Does 2019 Lee (doi: 10.20463/jenb.2019.0008) fit into your criteria? 

Why wasn't the 2018 Cuenca (doi: 10.3390/nu10091222) study included? The assessed fatigue. 

Can you report the effect sizes from the 15 studies you included? Since 60% showed positive results this may add to the paper. 

I would suggest finalizing a review of the pertinent literature and seeing if you missed anything. 

Author Response

Dear Editor and reviewers:

We have carefully considered all reviewers' recommendations for the paper (Manuscript ID: ijerph-1827791) entitled "Effectiveness of nitrate intake on recovery from exercise-related fatigue: a systematic review”. Please find enclosed our detailed answers to reviewers' queries. The authors declare that the manuscript is original and has not been considered for publication elsewhere. Additionally, the authors had approved the paper for release and agree with its content.

Please find all corrections in red inside the manuscript.

Reviewer 1

This is a systematic review of the effectiveness of nitrate supplementation on exercise-related fatigue. Overall the paper is well written and concise. The table was well thought-out and was easy to read. The discussion summarized the table and discussed the limitations, practical applications, and main points. This was a novel approach to reviewing nitrate supplementation. There are multiple reviews and meta-analyses reviewing performance. This seems like the first to review fatigue. 

R/ As Corresponding Author On Behalf Of All Authors We Thank The Reviewers For Their Contributions To Improve The Quality Of This MS.

Below are a few suggestions to strengthen the paper. 

Line 23: I don't think focusing on "organic" adds to the paper. Almost all supplementation is organic. This was not developed through the paper and I think it is assumed.

R/We really agree with the reviewer and we have deleted this sentences at the abstract and introduction.

When referencing the studies performed with performance in the intro you might want to reference the 2021 review & meta-analysis from MSSE (DOI: 10.1249/MSS.0000000000002363), it seemed like it fit your criteria. 

R/We want to thank the reviewer for his/her recommendation. This suggested MS fits our criteria and we have included.

You may want to consider mental and cognitive fatigue as well, 2014 Thompson (doi: 10.1016/j.resp.2013.12.015).

R/We have considered it in the introduction section but it should be excluded from the analysis due to the main objective of the MS is focused on performance.

Does 2019 Lee (doi: 10.20463/jenb.2019.0008) fit into your criteria? 

Why wasn't the 2018 Cuenca (doi: 10.3390/nu10091222) study included? The assessed fatigue. 

Although the manuscripts of Cuenca and Lee are related to fatigue, it does not assess recovery and the fatigue was evaluated during the sprints (in between). So, it did not fit our criteria. The reasons for the exclusion of the manuscripts were added in this section. This section of the methods was clarified.

Can you report the effect sizes from the 15 studies you included? Since 60% showed positive results this may add to the paper. 

effect sizes were not calculated because we considered including all variables related to recovery regardless of their type. The heterogeneity of variables is a factor that makes it difficult to interpret the effect sizes together. This was rescued in the limitations of the study.

I would suggest finalizing a review of the pertinent literature and seeing if you missed anything. 

R/We have reviewed the recently published literature and we did not miss a MS considering our criteria. We want to give all our appreciation to the reviewer for his/her time.

Reviewer 2 Report

The aim of this paper, according to the authors, is to summarize the available literature on the effect of NO3- consumption on exercise-related fatigue and muscle damage. This review revealed that more than 60% of the included investigations reported positive effects on recovery indicators due to NO3- intake. However, that no markers of exercise recovery were significantly. The authors conclude that due to the heterogeneity of consumption doses, subject characteristics and fatiguing stimuli and variables used, it is bold to indicate that there is real effectiveness of NO3- consumption in recovery in sports and exercise.

The manuscript presents an important contribution to the literature about the effects of NO3- consumption. The topic is very interesting and the paper is well written.

I have few specific comments:

The introduction outlines the background literature and justifies the current study.

In the Methods section, please clarify the inclusion criteria based on the date of the studies, from 2000 to 2021, line 92, or 2000 to 2022, line 104? Were there no studies found up to 2016 that met the inclusion criteria? Please explain.

In the results, should be corrected the mistakes in figure 1. “Studies excluded based on inclusión and exclusión criterio”.

Line 126. “The remaining 47 studies were examined in full text, of which 15 articles fulfilled the inclusion criteria.” The authors should add brief reasons to exclusion of these 32 articles.

Line 142 “, ,”

Line 143 and line 238. Please, add a space after bracket.

In the first paragraph of the discussion, the adverb "however" seems misplaced in that sentence.

Line 226. Add “.” at the end of paragraph.

Line 287. “Additionally, one must be very careful with prescribing these supplements for recovery in those populations that have not been studied, such as women, elite athletes, older adults, and people with pathologies, among others who exercise or do sports.” As exclusion criteria for this review, it is established that studies in people with some pre-existing pathology were not chosen. Therefore, in this statement, "and people with pathologies", should not be attached since the authors have expressly not chosen this type of study.

Line 292, NO3 and NO2 format must be corrected.

In general, if you need hyphenation, you should break the word at an obvious boundary, at syllable boundaries. Check the document and put a hyphen placed after one of the syllables that make up the word (e.g. lines 54, 270, 292, 327).

In Table 1, you need to align the table text to the left, if you justify the text it can cause confusion and certainly harming readability. Also, you will need to consider change and adjust spacing before and after paragraphs.

Author Response

Dear Editor and reviewers:

We have carefully considered all reviewers' recommendations for the paper (Manuscript ID: ijerph-1827791) entitled "Effectiveness of nitrate intake on recovery from exercise-related fatigue: a systematic review”. Please find enclosed our detailed answers to reviewers' queries. The authors declare that the manuscript is original and has not been considered for publication elsewhere. Additionally, the authors had approved the paper for release and agree with its content.

Please find all corrections in red inside the manuscript.

REVIEWER 2

The aim of this paper, according to the authors, is to summarize the available literature on the effect of NO3- consumption on exercise-related fatigue and muscle damage. This review revealed that more than 60% of the included investigations reported positive effects on recovery indicators due to NO3- intake. However, that no markers of exercise recovery were significantly. The authors conclude that due to the heterogeneity of consumption doses, subject characteristics and fatiguing stimuli and variables used, it is bold to indicate that there is real effectiveness of NO3- consumption in recovery in sports and exercise.

The manuscript presents an important contribution to the literature about the effects of NO3- consumption. The topic is very interesting and the paper is well written.

R/ As Corresponding Author On Behalf Of All Authors We Thank The Reviewers For Their Contributions To Improve The Quality Of This MS.

I have few specific comments:

The introduction outlines the background literature and justifies the current study.

R/Thank you for your observation.

In the Methods section, please clarify the inclusion criteria based on the date of the studies, from 2000 to 2021, line 92, or 2000 to 2022, line 104? Were there no studies found up to 2016 that met the inclusion criteria? Please explain.

R/We have corrected the years, should be read 2022. we based our decision to include manuscripts from 2000 to 2022 on other similar systematic reviews. we wanted to include all possible entries and thus identify the largest number of scientific articles. Regardless of the fact that the year of publication where the study of this topic apparently began is 2016.

In the results, should be corrected the mistakes in figure 1. “Studies excluded based on inclusión and exclusión criterio”.

R/the figure was corrected, thank you for highlight this isse.

Line 126. “The remaining 47 studies were examined in full text, of which 15 articles fulfilled the inclusion criteria.” The authors should add brief reasons to exclusion of these 32 articles.

R/the reasons for the exclusion of the manuscripts were added in this section.

Line 142 “, ,”
R/This typo was corrected.

Line 143 and line 238. Please, add a space after bracket.
The spaces were added.

In the first paragraph of the discussion, the adverb "however" seems misplaced in that sentence.

R/The adverb was corrected.

Line 226. Add “.” at the end of paragraph.
R/the dot was added.

Line 287. “Additionally, one must be very careful with prescribing these supplements for recovery in those populations that have not been studied, such as women, elite athletes, older adults, and people with pathologies, among others who exercise or do sports.” As exclusion criteria for this review, it is established that studies in people with some pre-existing pathology were not chosen. Therefore, in this statement, "and people with pathologies", should not be attached since the authors have expressly not chosen this type of study.

R/Thank you for highlighting this issue we agree and the sentence was deleted.